# In Silico Investigation of the Biological Implications of Complex DNA Damage with Emphasis in Cancer Radiotherapy through a Systems Biology Approach

**DOI:** 10.3390/molecules26247602

**Published:** 2021-12-15

**Authors:** Athanasia Pavlopoulou, Seyedehsadaf Asfa, Evangelos Gioukakis, Ifigeneia V. Mavragani, Zacharenia Nikitaki, Işıl Takan, Jean-Pierre Pouget, Lynn Harrison, Alexandros G. Georgakilas

**Affiliations:** 1Izmir Biomedicine and Genome Center, Balcova, Izmir 35340, Turkey; athanasia.pavlopoulou@ibg.edu.tr (A.P.); sadaf.asfa@msfr.ibg.edu.tr (S.A.); isil.takan@msfr.ibg.edu.tr (I.T.); 2Izmir International Biomedicine and Genome Institute, Genomics and Molecular Biotechnology Department, Dokuz Eylül University, Balcova, Izmir 35220, Turkey; 3Physics Department, School of Applied Mathematical and Physical Sciences, National Technical University of Athens (NTUA), 15780 Zografou, Greece; vgioukakhs@gmail.com (E.G.); ifimav@mail.ntua.gr (I.V.M.); znikitaki@mail.ntua.gr (Z.N.); 4IRCM, Institut de Recherche en Cancérologie de Montpellier, INSERM U1194, Université de Montpellier, Institut Régional du Cancer de Montpellier, 34298 Montpellier, France; jean-pierre.pouget@inserm.fr; 5Department of Molecular and Cellular Physiology, Louisiana State University Health Sciences Center, Shreveport, LA 71130, USA; lynn.clary@lsuhs.edu

**Keywords:** clustered DNA damage, ionizing radiation, cancer, comorbidities, prognosis, systems biology

## Abstract

Different types of DNA lesions forming in close vicinity, create clusters of damaged sites termed as “clustered/complex DNA damage” and they are considered to be a major challenge for DNA repair mechanisms resulting in significant repair delays and induction of genomic instability. Upon detection of DNA damage, the corresponding DNA damage response and repair (DDR/R) mechanisms are activated. The inability of cells to process clustered DNA lesions efficiently has a great impact on the normal function and survival of cells. If complex lesions are left unrepaired or misrepaired, they can lead to mutations and if persistent, they may lead to apoptotic cell death. In this in silico study, and through rigorous data mining, we have identified human genes that are activated upon complex DNA damage induction like in the case of ionizing radiation (IR) and beyond the standard DNA repair pathways, and are also involved in cancer pathways, by employing stringent bioinformatics and systems biology methodologies. Given that IR can cause repair resistant lesions within a short DNA segment (a few nm), thereby augmenting the hazardous and toxic effects of radiation, we also investigated the possible implication of the most biologically important of those genes in comorbid non-neoplastic diseases through network integration, as well as their potential for predicting survival in cancer patients.

## 1. Introduction

Many decades of research in radiobiology have shown that the inability of cells to repair DNA damage can be markedly detrimental, not only for the cells but for the whole organism, as well [1,2]. There are various types of DNA lesions that can be generated endogenously by metabolism or by exogenous agents [3]. These include nucleobase damage in the form of oxidation, deamination, alkylation or chlorination, breaks in one of the two strands of the DNA molecule, referred to as single-strand breaks (SSB), or breaks in both strands of the DNA, known as double-strand breaks (DSB) [4]. When different types of lesions form in close vicinity (within one or two helical turns of the DNA strand), they create clusters of damaged sites termed as “clustered/complex DNA damage” or “multiply damaged sites” and they are considered to be a major challenge for DNA repairing mechanisms [5]. Complex DNA lesions are commonly caused by ionizing radiation (IR), as low energy photons or charged particles traversing cells induce clusters of excitations and ionizations, causing lesions within a short DNA segment (a few nm), therefore augmenting the hazardous effects of radiation [6]. Such damage clusters are repair resistant, increasing genomic instability and malignant transformation, and can be considered as “danger” signals promoting chronic inflammatory response and leading to detrimental effects to the organism, such as radiation toxicity [7]. Other exogenous sources of DNA damage are chemicals and therapeutic drugs used in cancer treatment, like cisplatin- or carboplatin-alkylating agents. DNA lesions caused by endogenous factors like DNA replication and oxidative stress, in most cases, are evenly distributed along the DNA molecule and are easily repaired by the cell [4,8]. Our central hypothesis is that complex DNA damage induced by IR is the primary instigator of biological and clinical responses to cancer radiotherapy (RT). 

Normally, when DNA damage is induced, the corresponding DNA damage response and repair (DDR/R) mechanism is activated. DDR/R is a complex process consisting of recognition, signaling and repair of DNA damage [9]. Based on the lesion type, different DDR mechanisms are recruited. In the case of nucleobase damage, Base Excision Repair (BER) or Nucleotide Excision Repair (NER) is recruited, whereas in the case of DNA replication errors and other mismatch errors, Mismatch Repair (MMR) is initiated. SSBs are addressed with BER and NER as well, while DSBs (the most deleterious of DNA lesions) are processed by two different mechanisms namely, Non-Homologous End Joining (NHEJ) and the more accurate Homologous Recombination (HR), along with their sub-pathways [10,11]. The single DNA lesions are repaired efficiently and relatively easily. However, in the case of complex DNA damage, the recruitment of different DDR/R proteins to individual damages within the clustered lesions and their rapid processing is impeded, leading to considerable delays in the efficient damage repair. DNA repair deficiencies contribute largely to the increased mutagenicity of complex DNA damage. Accumulating evidence suggests that deficiencies in repair enzymes like DNA-PK and APE1 significantly increase the formation of clustered DNA lesions and genomic instability [12]. The inability of cells to process clustered DNA lesions efficiently, has a great impact on the normal function and survival of cells [13]. If complex lesions are left unrepaired or misrepaired, they can lead to mutations and/or chromosomal aberrations, such as deletions and inversions. Such complex genetic changes are leading indicators of genomic instability and potentially of carcinogenesis. In some cases, persistent clusters of damaged DNA can trigger cell death through apoptosis [7,14]. 

Rapid advances in high-throughput technologies in biology resulted in the accumulation of a great amount of biological data. This led to the emergence of the interdisciplinary field of Systems Biology, which could be defined as the application of computational methodologies for the integration, analysis and interpretation of diverse and heterogeneous biological data. Therefore, systems-based approaches enable a holistic insight into the investigation of complex biological problems at the cell, tissue, or organism level [15,16,17], including diseases [18,19], therapeutics [20], pharmacology [21], pathogen-host interactions [22], metabolism [23] etc. 

Herein, we have employed a systems biology approach to investigate the impact of clustered DNA lesions on different aspects of human biology. We have particularly focused on the identification of human genes that are activated upon induction of complex DNA damage and are also known to be involved in cancer-related pathways. To this end, we mined bioinformatic databases to retrieve all genes that have a reported connection with complex DNA damage. The genes already known to be implicated in DDR mechanisms were not included in this study. We also sought to detect the most biologically important of those genes (“hubs”), so as to investigate their possible implications in tumor radioresistance and clinical radiosensitivity, as well as their possible association with other diseases besides cancer. Furthermore, the prognostic potential of the hub genes in diverse types of cancers was investigated. The findings of this study could be utilized for the development of new clinical strategies to enhance, for example, the effectiveness of oncological radiation treatments or to predict the radiosensitivity of patients to IR.

## 2. Results and Discussion

### 2.1. Retrieval of Genes

The aim of this study was to find human genes that are explicitly activated upon induction of clustered/complex DNA damage (e.g., after exposure to IR), beyond the expected standard DNA repair mechanisms. Therefore, these genes should not be part of the five known DNA repair mechanisms (BER, NER, NHEJ, HR, MMR) and should be activated only when the cell detects complex lesions in the DNA.

In this way, 831 gene terms were obtained regarding the “cellular response to DNA damage” and 319 genes implicated in the five main DNA repair mechanisms were acquired after duplicate removal. After the exclusion of the common DDR genes retrieved (Section 3.1.1), a total of 532 genes were retained (Appendix A, Figure 1). For “clustered DNA damage”, 869 gene terms were found. Functional enrichment analysis of those genes was performed in order to detect the ones implicated in cancer-relevant pathways (Section 3.2). This reduced the number to 300 genes and after the removal of the DDR-relevant genes, a set of 258 genes were chosen (Appendix A, Figure 1). A total of 44 genes implicated both in “clustered DNA damage” and in “cellular response to DNA damage” were chosen for further processing, the products of which appear to form a highly connected network (Figure 1). These genes are distributed into major signalling pathways such as cell cycle *(ATR*, *CCNB1/D1*, *CDKN1A/1B*, *CHEK2*, *MCM7*, *MDM2*, *MYC*, *PLK1*) [24], apoptosis (*BAX*, *BCL2*, *BCL2L1*, *IKBKG*, *MCL1*, *PIK3R1*, *TNF*, *TNFRSF1A*, *XIAP*) [25,26,27] or inflammation (*HMOX1*, *TNF*, *TNFRSF1A*, and *BAK1*) [28,29,30], further highlighting their indispensable role in cellular physiology. Of note, the gene *IRF3* is also implicated in the cGAS-STING signalling pathway which is related to cellular response to cytosolic dsDNA [31]; in this case free cytosolic dsDNAs acts as endogenous danger signaling molecules or disease-associated molecular patterns (DAMP) [32] which can initiate downstream proinflammatory signaling events [33].

Moreover, 20 genes were found to be common to those reported in IntOGen, a comprehensive and integrated database of cancer driver genes and pathways [34] (Appendix A), further supporting a mechanistic connection between key genes related to IR response of complex DNA damage and cancer. Of note, a non-negligible overlap (20%) was found between the 44 genes and those genes found to be differentially expressed upon exposure of normal cells to α-radiation as compared to the non-irradiated ones (Appendix A). 

### 2.2. Associations among Genes, Cancers and Non-Neoplastic Diseases

Accumulating evidence points to an intimate link between cancer and comorbidity [35]. Furthermore, there are several eminent studies that highlight the impact of the acute or late effects of radiation-induced damage on human physiology. For example, Apollo lunar astronauts exposed to more intense deep space radiation exhibited higher mortality risk due to cardiovascular diseases as compared to astronauts in low Earth orbit [36]. Radiation also has a pronounced impact on the cancer patients undergoing RT [37].

Herein, gene-centered network diffusion [38] was applied for uncovering relationships among cancers, co-morbid diseases and radiotoxicity. The top nodes that correspond to the associations of the 44 genes, neoplasms and non-neoplastic diseases (Appendix A) were detected by combining the output of the eleven methods in cytoHubba (Figure 2) (Section 3.5). The most prominent neoplasms are the colorectal, lung, breast, prostatic and endocrine gland malignancies, as well as osteosarcoma. 

In the generated network, the antiapoptotic BCL2 [39] is linked to hypertension, diabetes mellitus and colon cancer (Figure 2). Of note, studies have shown that BCL2 regulates radioresistance in colon cancer cells [40] and is up-regulated in radioresistant breast cancer cells [41]. Higher incidence of cardiovascular problems has also been observed in breast cancer patients when irradiated on the left breast versus the right, which is related to the late effect of RT [42]. In a study by Duma et al., a positive correlation between blood glucose levels and the dose of RT in glioblastoma multiforme (GBM) patients was shown [43].

The growth factor EGFR, which is associated with several neoplasms, is also related to acute kidney injury (Figure 2). Radiation-induced nephropathy or kidney toxicity is a common side effect of RT targeting pelvic cancers, like prostatic or colorectal neoplasms [44,45] (Figure 2).

The node corresponding to the gene *HMOX1* (*heme oxygenase 1*) is the most highly connected to the non-neoplastic diseases (Figure 2). HMOX1 and TNF (tumor necrosis factor) are associated with several diseases of diverse tissue origin, quite a few of which are also suggested to be linked to the adverse effects of RT, like cardiovascular diseases [46,47], colitis [48], diseases related to blood glucose, status epilepticus, kidney [44,45] and liver [49] diseases (Figure 2). 

The pro-apoptotic BAX (BCL2-associated X protein) [50], was found to be down-regulated in acute lymphoblastic leukemia (ALL) and breast cancer cells resistant to radiation [41]. BAX is also linked to non-neoplastic diseases of different tissue origin (Figure 2), suggesting also a potential role of this molecule in radiotoxicity.

In a previous study of ours, HMOX1, TNF, TNFRSF1A, and BAX were shown to be implicated in the innate immunity and inflammatory response of irradiated human tissue. Given that inflammation is interconnected to RT-associated toxicity [37,51], the role of the DNA damage relevant genes in the immune-mediated adverse reactions to RT should be further explored [52]. Pollard and Gatti further support that many radiotoxicity cases have been associated with underlying DNA repair disorders like Falconi anemia [53].

### 2.3. Prognostic Value of DNA Damage-Associated Genes in Diverse Cancers

The expression level of the *BAX*, *BCL2*, *CCND1*, *HMOX1*, and *TNF* genes was found to be strongly correlated with clinical outcome endpoints of cancer patients. As anticipated, a statistically significant relationship of the elevated expression of the pro-apoptotic *BAX* and the tumor suppressor *TNF* [54] with favorable prognosis was observed for different types of cancers, as indicated by hazard ratio (HR) values < 1 and *p*-values < 0.05 (Figure 3 and Table 1). The overexpression of *BCL2* [55,56], *CCND1* [57,58], and *HMOX1* [59], which are either oncogenes or genes with oncogenic potential, was significantly associated with poor prognosis in multiple types of cancers, with HR > 1 (Figure 3 and Table 1).

For almost all types of IRs, the induction of dense ionizations and closely spaced DNA lesions are considered a signature for these types of radiations from low energy photons and electrons [60] to space radiations and a major contribution in the detrimental effects of IRs and instigator of biological responses at the cellular or tissue level. Furthermore, the genes *BAX*, *BCL2*, *CCND1*, and *TNF* were found to be differentially expressed in mammalian cancer cells exposed to medium-to-high-LET radiation versus non-irradiated cells. In particular, the genes *BAX* [61] and *TNF* [62] were found to be downregulated in proton-irradiated breast cancer cells; conversely, *BCL2* and *CCND1* were up-regulated in carbon-radiated human medulloblastoma cells [63]. This signifies the importance of these genes in cancer RT.

Highly dense biological damage is not necessarily restricted to DNA but also proteins and lipid membranes, but over the years, emphasis has been given to the DNA. The generation of systemic responses in bystander or distant cells are included in the major effects of complex biological damage with the involvement of innate immune system responses [64].

Therefore, based on data derived from a rather large cohort of cancer patients, DNA damage-relevant genes represent powerful prognostic cancer biomarkers. The predictive potential of these genes in radiotoxicity for the cancer patients undergoing RT should be taken into consideration, as the correlation between gene expression determined before RT and clinical response in patients can be used as a biomarker to identify radiosensitive individuals. This would be particularly useful in clinical decision-making before applying RT regarding the anticipation effect of this modality. For example, RT is expected to be more effective for patients with decreased levels of *BCL2* and increased expression of *BAX* as compared to fellow patients with opposite gene expression patterns [37,41]. Of note, cancer-associated comorbidities are associated with poor clinical outcomes in cancer patients [35,65]. Therefore, the full spectrum of comorbid diseases should be taken into consideration during cancer patients’ diagnosis, prognosis and treatment [66].

## 3. Materials and Methods

### 3.1. Collection and Selection of Candidate Genes

In order to identify human genes that are triggered after IR-generated DNA damage and are also involved in cancer related pathways, we performed extensive mining of relevant bioinformatics databases and resources, using bioinformatics tools: (i) the Gene Ontology (GO) (http://geneontology.org/; accessed on 10 March 2021) knowledgebase is a universal, comprehensive source of ontology concepts for describing fundamental characteristics of gene products across species, including biological process, molecular function and cellular component [67]; (ii) BioMart is an open-source community portal, based on a data federation framework, which provides a single, unified point of accessing and retrieving all available data [68,69]. Ensembl [70] is a genome database which hosts the BioMart community. Ensembl Genes mart contains well-annotated gene-centric data from diverse taxa [71]; (iii) The National Center for Biotechnology Information (NCBI) GenBank is a publicly available and comprehensive database of annotated nucleotide sequences across the tree of life [72,73]. 

Three sets of genes were compiled, as described below.

#### 3.1.1. DNA Repair Mechanisms

The Gene Ontology (GO) terms GO:0006284, GO:0006289, GO:0006298, GO:0035825, and GO:0070419 corresponding to the five major DNA repair mechanisms BER, NER, MMR, HR and NHEJ, respectively, were retrieved via the *AmiGO 2* web application [74]. 

The database Ensembl Genes 103 and the dataset ‘*Homo sapiens* genes (GRCh38.p13)’ were selected from the Ensembl Biomart [71] to retrieve those genes connected with the aforementioned GO terms. Each GO term was used separately as ‘Filter’. The ‘Features’, ’Gene name’ and ‘Gene description’ were chosen under ‘Attributes’, in order to retrieve five TXT files containing gene entries.

#### 3.1.2. Cellular Response to DNA Damage

The broad GO term “cellular response to DNA damage stimulus” (GO:0006974) was accessed through *AmiGO 2* [74]. This gene set was complemented with gene names which were retrieved by searching the ‘Gene’ database (https://www.ncbi.nlm.nih.gov/gene; accessed on 25 March 2021) of NCBI [72,73] by using the relevant keywords “response to DNA damage stimulus”, “DNA damage response”, “response to genotoxic stress” and “cellular DNA damage response”.

#### 3.1.3. Complex DNA Damage

To collect gene terms related to “Clustered DNA damage”, the NCBI ‘Gene’ database [72] was queried with the keywords “complex DNA damage”, “clustered DNA damage”, “complex DNA lesions” and “clustered DNA lesions”. 

### 3.2. Gene Set Enrichment Analysis

Gene set enrichment analysis (GSEA), is a powerful method for identifying sets of genes which are involved in common biological functions and pathways, and are significantly over-represented in a large dataset [75]. In this study, GSEA was conducted for functionally annotating the gene sets under study in the context of cancer. To this end, WebGestalt (WEB-based GEne SeT AnaLysis Toolkit) [76,77] was used to identify statistically significant over-represented cancer terms integrated in WikiPathways [78,79] in a given gene dataset; the hypergeometric distribution was applied and the threshold for the False Discovery Rate (FDR)-corrected *p*-value was set at 10^−3^.

### 3.3. Protein-Protein Interactions Networks

The associations among the gene/gene products under study were investigated and displayed through STRING version 11.5 [80] (https://string-db.org/; accessed on 10 June 2021), a database which collects and integrates protein-protein association data, either primary or predicted. The confidence interaction score was above 0.7 to enhance the stringency of the associations, and avoid the inclusion of false positive associations.

### 3.4. Gene-Disease Associations

The associations between the genes of interest and neoplasms, as well as non-neoplastic diseases (i.e., possible comorbidities), were retrieved from the DisGeNET [81,82], a knowledge base platform including the entire spectrum of human diseases; only associations with a confidence score above 0.5 were selected (Appendix A). 

### 3.5. Functional Networks

A network was created by combining the known associations between (i) genes and neoplasms, and (ii) genes and non-neoplastic diseases in order to predict and prioritize comorbid diseases that are potentially associated with the adverse effects of RT.

Taking into account that radiation is delivered locoregionally, the anatomical location or the organs at which the radiation dose is administered affects largely the pathogenesis and clinical manifestations of radiotoxicity [51]. Thus, the body site of cancers (Appendix A) was considered as an attribute in the neoplasm nodes in the network. The neoplasms were linked to their corresponding body locations/organs based on information derived from the Medical Subject Headings (MeSH) thesaurus [83] maintained in NCBI. In addition, MEDLINE/PubMed (https://pubmed.ncbi.nlm.nih.gov/; accessed on 20 June 2021) was searched thoroughly using keywords including a given “cancer type*” and “site”, “location”, “tissue”, “organ”. The 15 most relevant articles were carefully read to detect any associations between neoplasms and body sites or tissues.

The topological features of the generated network were investigated with the usage of Cytoscape (v.3.8.2) (https://cytoscape.org/; accessed on 23 July 2021) [84], a software platform for network processing, integration and visualization. Moreover, the Cytoscape plugin cytoHubba [85], which includes eleven local-based and global-based methods for ranking nodes by their topological properties, was utilized to select the top-20 nodes. 

### 3.6. Survival Analysis

The prognostic significance of the genes *BAX*, *BCL2*, *CCND1*, *HMOX1*, and *TNF* for diverse types of cancers was also explored. To investigate whether the expression levels of these genes are associated with the overall survival (i.e., a person is either alive or dead) of cancer patients, the online tool GEPIA (Gene Expression Profiling Interactive Analysis) [86,87] version 2 (http://gepia2.cancer-pku.cn/#index; accessed on 3 August 2021) was utilized. GEPIA2 retrieves and processes survival data from the integrated The Cancer Genome Atlas (TCGA) pan-cancer clinical data resource [88], which can be easily retrieved and visualized by users. The cancer patient cohorts were classified as high-risk and low-risk; the thresholds for high and low gene expression level patient groups were set at 75% and 25%, respectively.

### 3.7. Transcriptome Response of Cells to Medium-to-High LET Radiation

Clustered DNA lesions are mainly induced by medium-to-high-LET (Linear Energy Transfer) ionizing radiation, like α-particles, carbon ions and protons [8]. In our study, MEDLINE/PubMed (https://pubmed.ncbi.nlm.nih.gov/; accessed on 21 November 2021) was searched for scientific articles related to the response of both normal and cancer mammalian cells/tissues to medium-to-high-LET IR. Only those genes differentially expressed between the LET-irradiated compared to the non-irradiated cells/tissues with an absolute log2 fold change (FC) above 1.5 (|log2FC ≥ 1.5|) (alternatively, log2FC ≥ 1.5 and <0.67), and adjusted *p*-value less than 0.05 or *p*-value < 0.001, were considered.

## 4. Conclusions

In the present study, a systems biology approach and bioinformatics tools were utilized to investigate the biological effects of complex DNA damage by combining, analyzing and interpreting relevant data from diverse resources. The findings of this study could be extrapolated to other cancer sites where radiation is delivered. The results should also be taken into consideration for investigating the role of pivotal DNA damage-associated genes (*HMOX1*, *TNF*, *BAX*, etc.) in radiotoxicity, since the genetic mechanisms underlying RT-associated toxicity remain largely unelucidated. The growing knowledge on the importance of clustered/complex DNA damage extends beyond the widely accepted idea of clustered lesions being merely ‘signatures’ of ionizing radiations, especially medium-to-high LET like alpha-particles. Our work has been based on the main hypothesis that all these proteins involved in the ‘confrontation’ of clustered damage generate a variety of systemic effects and signals leading to either cancer or non-cancerous diseases as also identified herein (Figure 2 and Figure 3) and based on an earlier suggestion by Georgakilas and colleagues [7].

The candidate genes associated with radiation-induced damage could provide the foundation for the design of combinatorial therapeutic approaches, so as to enhance sensitization of cancer cells during RT and at the same time, minimize the radiotoxicity in the adjacent normal tissues. Therefore, blocking the antiapoptotic and poor prognostic marker *BCL2* and inducing the exogenous expression of the pro-apoptotic and favorable prognostic marker *BAX* in cancer tissues, could be considered in the development of antioncogenic therapies strategies. Last but not least, it is important to further explore the role(s) of key pathways (recovered also in this work), like the cGAS-STING, which is suggested to regulate multiple responses to radiation damage including but not limited to anti-tumor immune response, radiation-induced necrosis and radiation-induced fibrosis (reviewed in [69]). Of importance, this study could pave the way for experimental investigations.

## Figures and Tables

**Figure 1 molecules-26-07602-f001:**
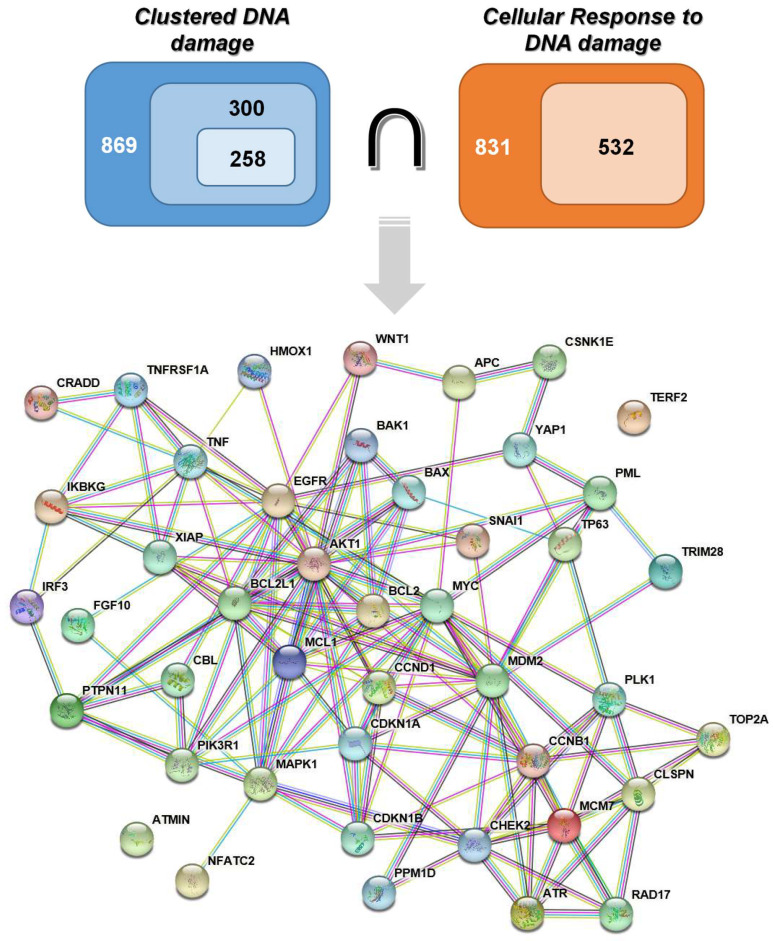
Workflow illustrating the selection of 44 genes associated with both clustered DNA damage and cellular response to DNA damage, after eliminating genes known to be involved in DDR.

**Figure 2 molecules-26-07602-f002:**
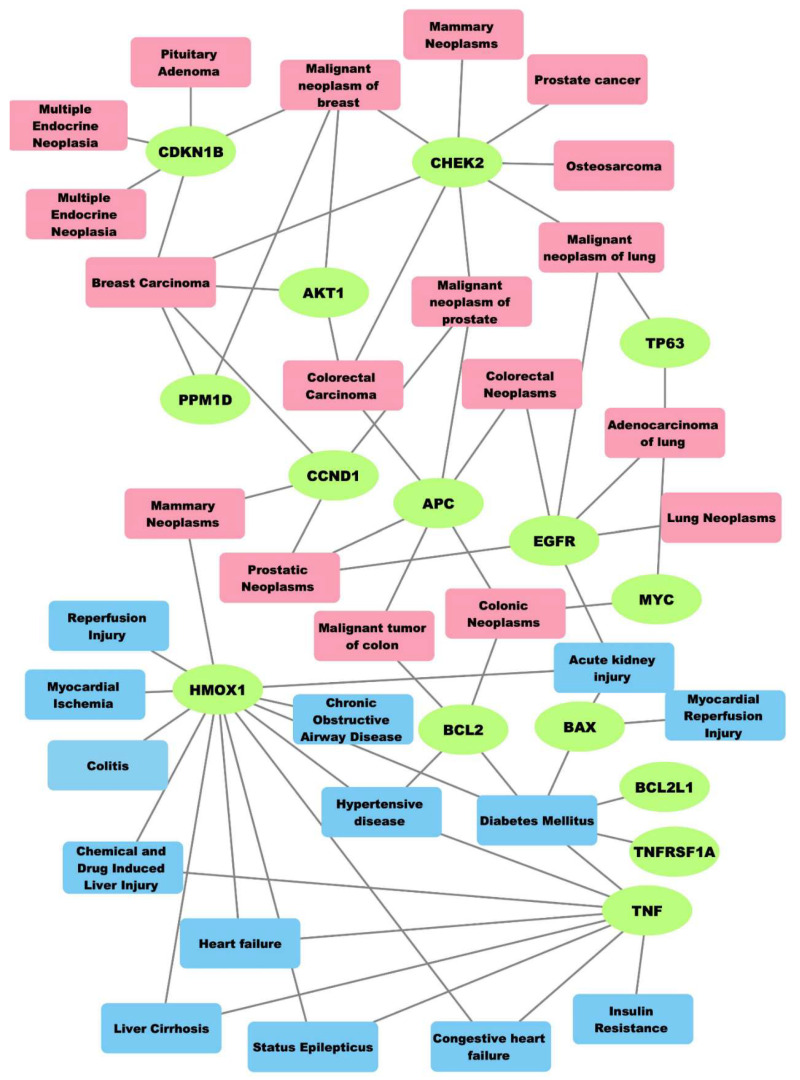
Integrated network depicting the top-ranking pairwise relationships (indicated by edges or lines) between genes-neoplasms, and genes and non-neoplastic diseases, derived from DisGeNET; green: genes, magenta: neoplasms, blue: non-neoplastic diseases often appearing as comorbitidies in various types of cancer therapies, like radiation therapies.

**Figure 3 molecules-26-07602-f003:**
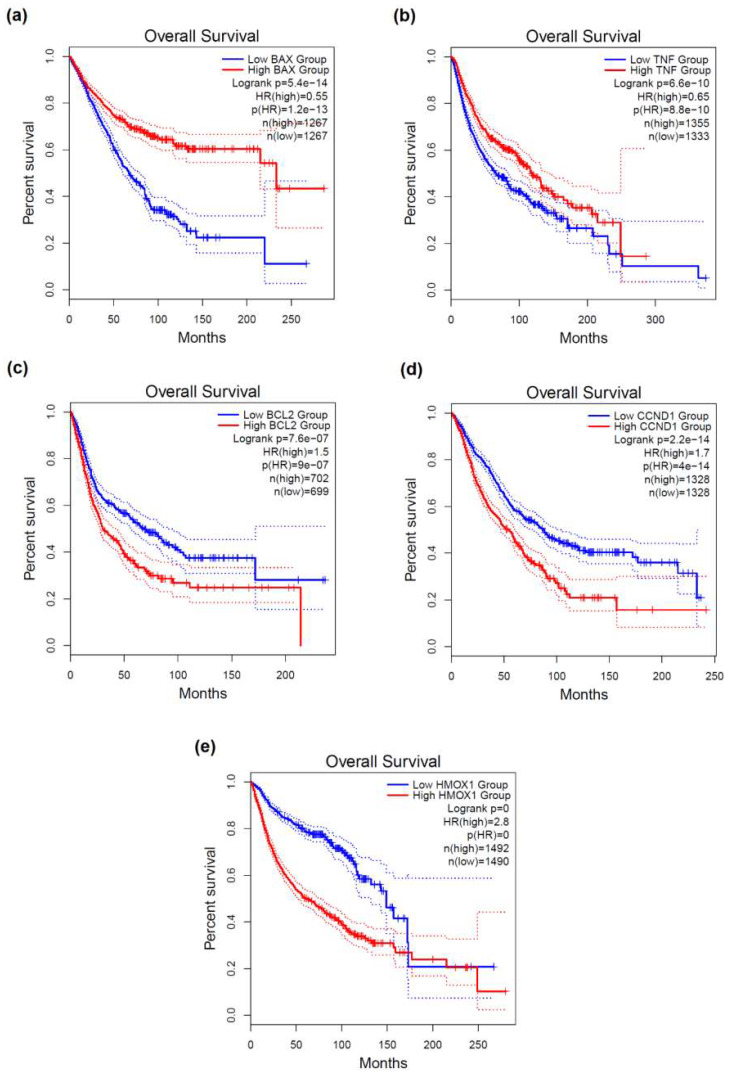
Kaplan–Meier curves representing the prognostic potential of selected pivotal genes (**a**) *BAX*, (**b**) *TNF*, (**c**) *BCL2*, (**d**) *CCND1*, and (**e**) *HMOX1* for overall survival in diverse cancers. The HR “HR(high)” and the corresponding *p*-values “p(HR)” are shown. The 95% confidence intervals (CI) are indicated by dotted lines. The number of high-risk and low-risk patient groups are denoted by “n(high)” and “n(low),” respectively. An HR value above 1 indicates an increased mortality risk, whereas an HR below 1 denotes a lower risk. Cancer patients with elevated expression of *BCL2*, *CCND1* and *HMOX1* may die at a higher rate per unit time of those where the genes *BAX* and *TNF* are overexpressed.

**Table 1 molecules-26-07602-t001:** List of TCGA cancer types and prognosis-associated with prognosis in the corresponding malignancy.

Cancer Type *	BAX	TNF	BCL2	CCND1	HMOX1
Adrenocortical carcinoma (ACC)	x	x	x	x	
Bladder urothelial carcinoma (BLCA)		x	x		x
Breast invasive carcinoma (BRCA)	x	x			x
Cervical squamous cell carcinoma and endocervical adenocarcinoma (CESC)		x		x	
Cholangio carcinoma (CHOL)	x		x	x	
Colon adenocarcinoma (COAD)	x		x	x	
Lymphoid neoplasm diffuse large B-cell lymphoma (DLBC)	x	x		x	x
Esophageal carcinoma (ESCA)	x	x		x	
Glioblastoma multiforme (GBM)			x		x
Head and neck squamous cell carcinoma (HNSC)			x	x	x
Kidney chromophobe (KICH)				x	x
Kidney renal clear cell carcinoma (KIRC)	x				
Kidney renal papillary cell carcinoma (KIRP)	x			x	
Acute myeloid leukemia (LAML)	x		x		
Brain lower grade glioma (LGG)		x			x
Liver hepatocellular carcinoma (LIHC)	x	x	x		x
Lung adenocarcinoma (LUAD)	x	x		x	x
Lung squamous cell carcinoma (LUSC)				x	x
Mesothelioma (MESO)		x	x		x
Ovarian serous cystadenocarcinoma (OV)	x			x	
Pancreatic adenocarcinoma (PAAD)		x	x	x	
Pheochromocytoma and paraganglioma (PCPG)	x			x	x
Prostate adenocarcinoma (PRAD)	x			x	x
Rectum adenocarcinoma (READ)		x	x	x	x
Sarcoma (SARC)	x	x		x	
Skin cutaneous melanoma (SKCM)		x			
Stomach adenocarcinoma (STAD)		x	x	x	
Testicular germ cell tumors (TGCT)	x	x	x	x	
Thyroid carcinoma (THCA)	x				x
Thymoma (THYM)	x	x		x	x
Uterine corpus endometrial carcinoma (UCEC)	x	x		x	x
Uterine carcinosarcoma (UCS)	x	x		x	x
Uveal melanoma (UVM)	x	x		x	x

* The Cancer Genome Atlas.

## Data Availability

Data will be available from the corresponding author upon logical request.

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
