# Peer review of "In Silico Investigation of the Biological Implications of Complex DNA Damage with Emphasis in Cancer Radiotherapy through a Systems Biology Approach"

_molecules, 2021, doi:10.3390/molecules26247602_

Round 1
Reviewer 1 Report
The manuscript “In silico investigation of the biological implications of complex DNA damage with emphasis in cancer radiotherapy through a systems biology approach” exploits bioinformatic tools to identify genes that are activated by clustered DNA damages, as such induced by high LET radiation, and that are involved in cancer-related pathways or in the onset of non-neoplastic diseases.
This kind of study provides the proof of principle for a methodology to be used in clinics to enhance the effectiveness of oncological radiation treatments or to predict the radio-sensitivity of patients to ionizing radiation (IR). Given the amount of data in the literature, these bioinformatics analyses help in discriminating among the plethora of possible biomarkers, that can be used in the prognosis of the disease.
The work could have been improved by the addition of experimental data on the effective role of the identified genes at the cellular level after IR in wild type cells (as standard for healthy patients), and in cells where the genes are over-expressed or knocked-out.
In any case, this analysis could pave the way for experimental investigations, and this should be underlined in the conclusions of the paper.
A point that could be improved is the description of the bioinformatic tools used in the manuscript, to make it accessible to all kinds of readership.
Moreover, it is not clear where a systems biology approach has been applied, the authors have to explain it better in the main text of the manuscript.
To make the manuscript more appealing, the authors could explain more exhaustively in the introduction and in the conclusions how and why bioinformatics and systems biology approaches can enrich the research/medical field, and the applications of these tools. The introduction especially, as it is right now, does not underline the novelty of the studied approach.
Author Response
RESPONSE TO REVIEWS
We thank the Editor and the reviewers for the constructive criticism and reviews. Our main effort was to revise the manuscript according to the comments and suggestions by reviewers. This is an in silico research study. Our main aim has been to identify the genes/proteins participating in the ‘confrontation’ of clustered DNA damage and lesions.
This was the starting point of our study and nothing else. Our methodologies were stringent and on the frontier of the bioinformatics field. We mined these genes out of thousands of studies and this is an original and unique approach not applied before, to the best of our knowledge. In the revision, we have compared with genes identified for induction of complex DNA damage by high-LET radiations (a genuine induces of clustered damage; please see revised Table S1: COMMON GENES with alpha particles). Other revisions have been made to make clearer and more precise our results and conclusions always in response to reviewers’ comments.
REVIEWER 1
The manuscript “In silico investigation of the biological implications of complex DNA damage with emphasis in cancer radiotherapy through a systems biology approach” exploits bioinformatic tools to identify genes that are activated by clustered DNA damages, as such induced by high LET radiation, and that are involved in cancer-related pathways or in the onset of non-neoplastic diseases.
This kind of study provides the proof of principle for a methodology to be used in clinics to enhance the effectiveness of oncological radiation treatments or to predict the radio-sensitivity of patients to ionizing radiation (IR). Given the amount of data in the literature, these bioinformatics analyses help in discriminating among the plethora of possible biomarkers, that can be used in the prognosis of the disease.
The work could have been improved by the addition of experimental data on the effective role of the identified genes at the cellular level after IR in wild type cells (as standard for healthy patients), and in cells where the genes are over-expressed or knocked-out.
We really appreciate the reviewer’s positive assessment of our work. In our study, we have conducted an in-depth hardcore in silico study, based on experimentally and clinically validated data (like gene-disease associations and patient survival data) derived from diverse sources. However, in the revised manuscript, we have further verified our identified 44 genes by comparing them to genes that are differentially expressed in mammalian cells (both normal and cancer) after exposure to medium or high LET radiations as genuine instigators of complex/clustered DNA damage) versus the corresponding non-irradiated cells (control). Relevant information was extracted from scientific articles in PubMed/MEDLINE, as described in Section 3.7.
- Hada, M.; Georgakilas, A.G. Formation of clustered DNA Damage after high-LET irradiation: A review. J. Radiat. Res. 2008, 49, 203-210.
In any case, this analysis could pave the way for experimental investigations, and this should be underlined in the conclusions of the paper.
We thank the reviewer for this comment. This indeed is our main aim, and we have underlined it in the conclusion of the paper.
A point that could be improved is the description of the bioinformatic tools used in the manuscript, to make it accessible to all kinds of readership.
We have made a great effort to describe the bioinformatics tools/databases in the revised version of our paper.
Moreover, it is not clear where a systems biology approach has been applied, the authors have to explain it better in the main text of the manuscript.
To make the manuscript more appealing, the authors could explain more exhaustively in the introduction and in the conclusions how and why bioinformatics and systems biology approaches can enrich the research/medical field, and the applications of these tools. The introduction especially, as it is right now, does not underline the novelty of the studied approach.
In the Introduction of the manuscript, we have added a paragraph describing the applicability of computational systems biology in experimental and medical research. In the conclusions, also, and throughout the manuscript, we have added text relevant to systems biology and the usage of bioinformatics tools.

Reviewer 2 Report
In the presented manuscript, the Authors use existing databases and bioinformatic tools to explore possibly unknown relationships between clustered lesions and non-neoplastic/non-cancerous comorbidity by identifying a subset of genes involved in DDR/R response but not directly in the major DNA repair mechanisms.
While the idea is promising, the data presented here doesn't take us very far exploring it - in fact, it barely scratches the surface of the problem. Since the authors only follow GO terms and broad associations with common diseases, it's impossible to tell if the connections identified are indeed mechanical, or just byproducts of general mechanisms like inflammation or apoptosis, as the Authors themselves point out - in which case the putative connection to clustered lesions would be strictly an artifact. On the other hand, the Intro exclusively focuses on the molecular aspect of clustered lesions, while no mechanistic insights related to this level of description are found in the paper. Figures are not very appealing and mostly convey little information on their own, as if they were screenshots from GUIs/web apps where the interpretation was left to the reader (they are discussed in the text, but in general, manuscript figures should be as self-explanatory as possible).
The K-M curves presented in Fig 3 also present little evidence for the putative connections - for example, the mortality isn't explicitly attributed to the comorbidities identified in Fig 2. In terms of improved prognostics/biomarker design, one could perform the survival analysis on the whole dataset, identifying genes crucial for cancer prognosis without referring to the question of clustered lesions. To me, this analysis lacks context (how do these genes fare against other potential biomarkers?), detail (how are we sure these are the actual comorbidities at play?) and a mechanical connection (how do we know clustered lesions were indeed involved here?).
Overall, I think this report is a promising introduction to an actual study on this no doubt interesting topic, but would benefit from a more cohesive and thoughtful design, as well as more precision/rigor and a better context.
Author Response
RESPONSE TO REVIEWS
We thank the Editor and the reviewers for the constructive criticism and reviews. Our main effort was to revise the manuscript according to the comments and suggestions by reviewers. This is an in silico research study. Our main aim has been to identify the genes/proteins participating in the ‘confrontation’ of clustered DNA damage and lesions.
This was the starting point of our study and nothing else. Our methodologies were stringent and on the frontier of the bioinformatics field. We mined these genes out of thousands of studies and this is an original and unique approach not applied before, to the best of our knowledge. In the revision, we have compared with genes identified for induction of complex DNA damage by high-LET radiations (a genuine induces of clustered damage; please see revised Table S1: COMMON GENES with alpha particles). Other revisions have been made to make clearer and more precise our results and conclusions always in response to reviewers’ comments.
REVIEWER 2
In the presented manuscript, the Authors use existing databases and bioinformatic tools to explore possibly unknown relationships between clustered lesions and non-neoplastic/non-cancerous comorbidity by identifying a subset of genes involved in DDR/R response but not directly in the major DNA repair mechanisms.
While the idea is promising, the data presented here doesn't take us very far exploring it - in fact, it barely scratches the surface of the problem. Since the authors only follow GO terms and broad associations with common diseases, it's impossible to tell if the connections identified are indeed mechanical, or just byproducts of general mechanisms like inflammation or apoptosis, as the Authors themselves point out - in which case the putative connection to clustered lesions would be strictly an artifact.
We thank the reviewer for his/her comments and suggestions. We feel though that we need to explain better our philosophy and idea and how all this has study been executed.
This in silico study is based on the processing, analysis and interpretation of data derived from diverse international, comprehensive, databases/consortia/web servers (e.g., NCBI, Gene Ontology consortium, GEPIA etc.) with the restriction of using specific keywords (i.e. relative to complex/clustered DNA damage) as explained analytically in the methods sections. The latter ones include only experimentally verified data from diverse sources. Therefore, the data analyzed in this study can be considered inclusive, complete and accurate, which could cover a broad spectrum of molecules, diseases, biological processes, protein-protein associations etc. We have used systems biology methods and state-of-the-art bioinformatics tools/resources to analyze relevant, publicly available, data.
In particular, we illustrate the utility of networks by applying them to combine known associations among cancers-genes-non-neoplastic diseases, in order to predict and prioritize comorbid diseases that are potentially associated with the adverse effects of RT; in other words, network-based methods can be used effectively and reliably to extrapolate information based on documented data. We believe that, this is a unique and original feature of our work in the broad radiation biology and DNA damage induction field.
On the other hand, the Intro exclusively focuses on the molecular aspect of clustered lesions, while no mechanistic insights related to this level of description are found in the paper.
We understand the essence of this comment and please mark that it was always from the very beginning also our main aim to provide mechanistic insights but within limitations of an in silico study.
In the revised version of the manuscript, we have included experimentally validated data on the effective role of the 44 identified genes at the cellular level after exposure to radiation. To this end, we investigated whether these 44 genes are differentially expressed in mammalian cells (both normal and cancer) after exposure to medium or high LET radiation (known to generate almost exclusively complex DNA damage; see below some suggested references) versus to their corresponding control, non-irradiated cells, based on published scientific studies (see Section 3.7).
- Hada, M.; Georgakilas, A.G. Formation of clustered DNA Damage after high-LET irradiation: A review. J. Radiat. Res. 2008, 49, 203-210.
Figures are not very appealing and mostly convey little information on their own, as if they were screenshots from GUIs/web apps where the interpretation was left to the reader (they are discussed in the text, but in general, manuscript figures should be as self-explanatory as possible).
In order to avoid any confusion, the figures in the manuscript are NOT screenshots, but they were copied/pasted from their original output files; these files can be provided upon request. Moreover, in the revised manuscript, Figure 2 has been replaced with one of higher resolution. In addition, the legend in Figure 3 has been revised in a way to be more self-explanatory.
The K-M curves presented in Fig 3 also present little evidence for the putative connections - for example, the mortality isn't explicitly attributed to the comorbidities identified in Fig 2.
We would like to explain this point better. Indeed, in many types of cancer therapies - actually in all types, non-neoplastic diseases appear as comorbidities but patient survival cannot be exclusively associated with these comorbidities. Therefore, we restrain from mentioning anywhere in the article that mortality is explicitly attributed to comorbid diseases; comorbidities can either represent contributing factors to carcinogenesis or outcome of cancer, since oncogenesis is a multifactorial process.
Based on clinical evidence (cited in the manuscript), a high prevalence of comorbidities are observed among cancer patients after treatment; there is a strong association between cancer and comorbid diseases, and consequently shorter overall survival (OS). Therefore, comorbidity should be taken into consideration in clinical decision-making before administering any treatment to the patient.
In our study, the clustered DNA damage-associated genes BAX, BCL2, CCND1, HOMX1 and TNF, which are also tightly linked to cancer, non-neoplastic diseases and radiotoxicity (Figure 2 and relevant text in the main manuscript), were shown to be powerful prognostic markers for OS in diverse cancers; over-expression of the anti-apoptotic BAX and the tumor suppressor TNF is significantly associated with favorable OS in diverse cancers, whereas elevated expression of the BCL2, CCND1 and HOMX1 (i.e. oncogenes or genes with oncogenic potential) is strongly correlated with poor prognosis in multiple types of cancers. Of importance, the genes BAX, BCL2, CCND1 and TNF, were also found to be differentially expressed in cancer cells exposed to medium-to-high-LET radiation versus non-irradiated cells, further signifying the importance of these genes in cancer radiotherapy.
In terms of improved prognostics/biomarker design, one could perform the survival analysis on the whole dataset, identifying genes crucial for cancer prognosis without referring to the question of clustered lesions.
Indeed, there are myriads of genes that could potentially serve as prognostic cancer biomarkers according to The Cancer Genome Atlas (TCGA) (https://tcga-data.nci.nih.gov) pan-cancer clinical data resource, which includes clinical data key features form more than 11,000 human tumors across 33 different cancer types.
In our study, one of our primary aims, taking also into consideration what the reviewer suggests above, was to provide meaningful results through rigorous bioinformatics analyses, which could discriminate among a plethora of possible biomarkers that can be used in the prognosis of the disease.
We are more than eager to crystallize all this and suggest five genes associated with clustered DNA damage that are significantly correlated to an important clinical outcome endpoint of cancer patients (i.e. overall survival).
To me, this analysis lacks context (how do these genes fare against other potential biomarkers?), detail (how are we sure these are the actual comorbidities at play?) and a mechanical connection (how do we know clustered lesions were indeed involved here?).
This analysis was conducted within the context of complex/clustered DNA damage. The corresponding author (Dr. Georgakilas) and co-author Dr. Lynn Harrison have devoted many years of research in this field of clustered DNA lesions induction and processing. The idea was based on the known and continuously advancing idea on the importance of these lesions not only as signatures of ionizing radiations especially medium-to-high LET but also for their systemic effects. We believe that all these proteins implicated in the initial recognition and processing of clustered DNA lesions confront them as a genuine type of stress beyond the DNA damage and repair process (DDR).
Please see our already cited references:
- Mavragani, I.V.; Nikitaki, Z.; Souli, M.P.; Aziz, A.; Nowsheen, S.; Aziz, K.; Rogakou, E.; Georgakilas, A.G. Complex DNA Damage: A Route to Radiation-Induced Genomic Instability and Carcinogenesis. Cancers 2017, 9, doi:10.3390/cancers9070091.
- Mavragani, I.V.; Nikitaki, Z.; Kalospyros, S.A.; Georgakilas, A.G. Ionizing Radiation and Complex DNA Damage: From Prediction to Detection Challenges and Biological Significance. Cancers (Basel) 2019, 11, doi:10.3390/cancers11111789.
Our aim was to highlight and shed light to the importance of genes associated with complex DNA damage in the clinical outcomes of diverse cancers probably through also systemic effects, since clustered lesions are known to contribute largely to carcinogenesis and systemic effects have not been investigated analytically.
Based on publicly available experimental evidence, those genes that have a potential prognostic role in multiple cancers were also found to be differentially expressed in cancer cells upon exposure to medium-to-high LET radiation which causes clustered lesions. This information is included in the revised manuscript, and provides further support to a mechanical connection between clustered damage-induced gene expression and cancer.
Overall, I think this report is a promising introduction to an actual study on this no doubt interesting topic, but would benefit from a more cohesive and thoughtful design, as well as more precision/rigor and a better context.
We respectfully ask the reviewer to read all our responses and comprehend that our work advances the knowledge on the field. Herein, we have conducted a hardcore in silico study based on experimentally validated data. We have revised the manuscript by taking into consideration the reviewers’ suggestions, so as to improve the quality of our work.
As we mention in the Conclusion, this analysis could pave the way for experimental investigations. The idea is considered original to our knowledge and thanks to both reviewers’ helpful comments we believe that our work has been advanced and been made more inclusive and precise.
